# Optimal Binary Autoencoding with Pairwise Correlations

**Akshay Balsubramani** [*]
Stanford University
abalsubr@stanford.edu

## Abstract

We formulate learning of a binary autoencoder as a biconvex optimization problem which learns from the pairwise correlations between encoded and decoded bits. Among all possible algorithms that use this information, ours finds the autoencoder that reconstructs its inputs with worst-case optimal loss. The optimal decoder is a single layer of artificial neurons, emerging entirely from the minimax loss minimization, and with weights learned by convex optimization. All this is reflected in competitive experimental results, demonstrating that binary autoencoding can be done efficiently by conveying information in pairwise correlations in an optimal fashion.

## 1 Introduction

Consider a general autoencoding scenario, in which an algorithm learns a compression scheme for independently, identically distributed (i.i.d.) $V$-dimensional bit vector data $\{\hat{\mathbf{x}}^{(1)}, \dots, \hat{\mathbf{x}}^{(n)}\}$. For some encoding dimension $H$, the algorithm encodes each data example $\hat{\mathbf{x}}^{(i)} = (\hat{x}_1^{(i)}, \dots, \hat{x}_V^{(i)})^\top$ into an $H$-dimensional representation $\mathbf{e}^{(i)}$, with $H < V$. It then decodes each $\mathbf{e}^{(i)}$ back into a reconstructed example $\tilde{\mathbf{x}}^{(i)}$ using some small amount of additional memory, and is evaluated on the quality of the reconstruction by the cross-entropy loss commonly used to compare bit vectors. A good autoencoder learns to compress the data into $H$ bits so as to reconstruct it with low loss.

When the loss is squared reconstruction error and the goal is to compress data in $\mathbb{R}^V$ to $\mathbb{R}^H$, this is often accomplished with principal component analysis (PCA), which projects the input data on the top $H$ eigenvectors of their covariance matrix (Bourlard & Kamp (1988); Baldi & Hornik (1989)). These eigenvectors in $\mathbb{R}^V$ constitute $VH$ real values of additional memory needed to decode the compressed data in $\mathbb{R}^H$ back to the reconstructions in $\mathbb{R}^V$, which are linear combinations of the eigenvectors. Crucially, this total additional memory does not depend on the amount of data $n$, making it applicable when data are abundant.

This paper considers a similar problem, except using bit-vector data and the cross-entropy reconstruction loss. Since we are compressing samples of i.i.d. $V$-bit data into $H$-bit encodings, a natural approach is to remember the pairwise statistics: the $VH$ average *correlations* between pairs of bits in the encoding and decoding, constituting as much additional memory as the eigenvectors used in PCA. The decoder uses these along with the $H$-bit encoded data, to produce $V$-bit reconstructions.

We show how to efficiently learn the autoencoder with the worst-case optimal loss in this scenario, without any further assumptions, parametric or otherwise. It has some striking properties.

The decoding function is identical in form to the one used in a standard binary autoencoder with one hidden layer (Bengio et al. (2013a)) and cross-entropy reconstruction loss. Specifically, each bit $v$ of the decoding is the output of a logistic sigmoid artificial neuron of the encoded bits, with some learned weights $\mathbf{w}_v \in \mathbb{R}^H$. This form emerges as the uniquely optimal decoding function, and is *not* assumed as part of any explicit model.

We show that the worst-case optimal reconstruction loss suffered by the autoencoder is convex in these decoding weights $\mathbf{W} = \{\mathbf{w}_v\}_{v \in [V]}$, and in the encoded representations $\mathbf{E}$. Though it is not

---

[*]Most of the work was done as a PhD student at UC San Diego.

jointly convex in both, the situation still admits a natural and efficient optimization algorithm in which the loss is alternately minimized in $\mathbf{E}$ and $\mathbf{W}$ while the other is held fixed. The algorithm is practical and performs well empirically, learning incrementally from minibatches of data in a stochastic optimization setting.

## 1.1 Notation

The observed data and encodings can be written in matrix form, representing bits as $\pm 1$:

$$\hat{\mathbf{X}} = \begin{pmatrix} \hat{x}_1^{(1)} & \cdots & \hat{x}_1^{(n)} \\ \vdots & \ddots & \vdots \\ \hat{x}_V^{(1)} & \cdots & \hat{x}_V^{(n)} \end{pmatrix} \in [-1,1]^{V \times n} \quad , \quad \mathbf{E} = \begin{pmatrix} e_1^{(1)} & \cdots & e_1^{(n)} \\ \vdots & \ddots & \vdots \\ e_H^{(1)} & \cdots & e_H^{(n)} \end{pmatrix} \in [-1,1]^{H \times n} \quad (1)$$

Here the encodings are allowed to be randomized, represented by values in $[-1,1]$ instead of just the two values $\{-1,1\}$; e.g. $e_i^{(1)} = \frac{1}{2}$ is $+1$ w.p. $\frac{3}{4}$ and $-1$ w.p. $\frac{1}{4}$. The data in $\mathbf{X}$ are also allowed to be randomized, which we will see essentially loses no generality (Appendix B). We write the columns of $\hat{\mathbf{X}}, \mathbf{E}$ as $\hat{\mathbf{x}}^{(i)}, \mathbf{e}^{(i)}$ for $i \in [n]$ (where $[s] := \{1, \ldots, s\}$), representing the data. The rows are written as $\hat{\mathbf{x}}_v = (x_v^{(1)}, \ldots, x_v^{(n)})^\top$ for $v \in [V]$ and $\mathbf{e}_h = (e_h^{(1)}, \ldots, e_h^{(n)})^\top$ for $h \in [H]$.

We also consider the correlation of each bit $h$ of the encoding with each decoded bit $v$ over the data, i.e. $b_{v,h} := \frac{1}{n} \sum_{i=1}^n x_v^{(i)} e_h^{(i)}$. This too can be written in matrix form as $\mathbf{B} := \frac{1}{n} \hat{\mathbf{X}} \mathbf{E}^\top \in \mathbb{R}^{V \times H}$, whose rows and columns we respectively write as $\mathbf{b}_v = (b_{v,1}, \ldots, b_{v,H})^\top$ over $v \in [V]$ and $\mathbf{b}_h = (b_{1,h}, \ldots, b_{V,h})^\top$ over $h \in [H]$; the indexing will be clear from context.

As alluded to earlier, the loss incurred on any example $\mathbf{x}^{(i)}$ is the cross-entropy between the example and its reconstruction $\tilde{\mathbf{x}}^{(i)}$, in expectation over the randomness in $\mathbf{x}^{(i)}$. Defining $\ell_\pm(\tilde{x}_v^{(i)}) = \ln\left(\frac{2}{1 \pm \tilde{x}_v^{(i)}}\right)$ (the *partial losses* to true labels $\pm 1$), the loss is written as:

$$\ell(\mathbf{x}^{(i)}, \tilde{\mathbf{x}}^{(i)}) := \sum_{v=1}^V \left[ \left(\frac{1+x_v^{(i)}}{2}\right) \ell_+(\tilde{x}_v^{(i)}) + \left(\frac{1-x_v^{(i)}}{2}\right) \ell_-(\tilde{x}_v^{(i)}) \right] \quad (2)$$

In addition, define a *potential well* $\Psi(m) := \ln(1 + e^m) + \ln(1 + e^{-m})$ with derivative $\Psi'(m) := \frac{1-e^{-m}}{1+e^{-m}}$. Univariate functions like this are applied componentwise to matrices in this paper.

## 1.2 Problem Setup

With these definitions, the autoencoding problem we address can be precisely stated as two tasks, encoding and decoding. These share only the side information $\mathbf{B}$. Our goal is to perform these steps so as to achieve the best possible guarantee on reconstruction loss, with no further assumptions. This can be written as a zero-sum game of an autoencoding algorithm seeking to minimize loss against an adversary, by playing encodings and reconstructions:

- Using $\hat{\mathbf{X}}$, algorithm plays (randomized) encodings $\mathbf{E}$, resulting in pairwise correlations $\mathbf{B}$.
- Using $\mathbf{E}$ and $\mathbf{B}$, algorithm plays reconstructions $\tilde{\mathbf{X}} = (\tilde{\mathbf{x}}^{(1)}; \ldots; \tilde{\mathbf{x}}^{(n)}) \in [-1,1]^{V \times n}$.
- Given $\tilde{\mathbf{X}}, \mathbf{E}, \mathbf{B}$, adversary plays $\mathbf{X} \in [-1,1]^{V \times n}$ to maximize reconstruction loss $\frac{1}{n} \sum_{i=1}^n \ell(\mathbf{x}^{(i)}, \tilde{\mathbf{x}}^{(i)})$.

To incur low loss, the algorithm must use an $\mathbf{E}$ and $\mathbf{B}$ such that no adversary playing $\mathbf{X}$ can inflict higher loss. The algorithm never sees $\mathbf{X}$, which represents the worst the data could be given the algorithm's incomplete memory of it $(\mathbf{E}, \mathbf{B})$ and reconstructions $(\tilde{\mathbf{X}})$.

We find the autoencoding algorithm's best strategy in two parts. First, we find the optimal decoding function of any encodings $\mathbf{E}$ given $\mathbf{B}$, in Section 2. Then, we use the resulting optimal reconstruction function to outline the best encoding procedure, i.e. one that finds the $\mathbf{E}, \mathbf{B}$ that lead to the best reconstruction, in Section 3.1. Combining these ideas yields an autoencoding algorithm in Section

3.2 (Algorithm 1), where its implementation and interpretation are specified. Further discussion and related work in Section 4 are followed by more extensions of the framework in Section 5. Experiments in Section 6 show extremely competitive results with equivalent fully-connected autoencoders trained with backpropagation.

## 2 OPTIMALLY DECODING AN ENCODED REPRESENTATION

To address the game of Section 1.2, we first assume $\mathbf{E}$ and $\mathbf{B}$ are fixed, and derive the optimal decoding rule given this information. We show in this section that the form of this optimal decoder is precisely the same as in a classical autoencoder: having learned a weight vector $\mathbf{w}_v \in \mathbb{R}^H$ for each $v \in [V]$, the $v^{th}$ bit of each reconstruction $\tilde{\mathbf{x}}^i$ is expressed as a logistic function of a $\mathbf{w}_v$-weighted combination of the $H$ encoded bits $\mathbf{e}^i$ – a logistic artificial neuron with weights $\mathbf{w}_v$. The weight vectors are learned by convex optimization, despite the nonconvexity of the transfer functions.

To develop this, we minimize the worst-case reconstruction error, where $\mathbf{X}$ is constrained by our prior knowledge that $\mathbf{B} = \frac{1}{n}\mathbf{X}\mathbf{E}^\top$, i.e. $\frac{1}{n}\mathbf{E}\mathbf{x}_v = \mathbf{b}_v \ \forall v \in [V]$. This can be written as a function of $\mathbf{E}$:

$$\mathcal{L}_{\mathbf{B}}^*(\mathbf{E}) := \min_{\tilde{\mathbf{x}}^{(1)},\ldots,\tilde{\mathbf{x}}^{(n)} \in [-1,1]^V} \ \max_{\substack{\mathbf{x}^{(1)},\ldots,\mathbf{x}^{(n)} \in [-1,1]^V, \\ \forall v \in [V]: \ \frac{1}{n}\mathbf{E}\mathbf{x}_v = \mathbf{b}_v}} \ \frac{1}{n} \sum_{i=1}^{n} \ell(\mathbf{x}^{(i)}, \tilde{\mathbf{x}}^{(i)}) \tag{3}$$

We solve this minimax problem for the optimal reconstructions played by the minimizing player in (3), written as $\tilde{\mathbf{x}}^{(1)*}, \ldots, \tilde{\mathbf{x}}^{(n)*}$.

**Theorem 1.** *Define the* bitwise slack function $\gamma^{\mathbf{E}}(\mathbf{w}, \mathbf{b}) := -\mathbf{b}^\top\mathbf{w} + \frac{1}{n}\sum_{i=1}^{n}\Psi(\mathbf{w}^\top\mathbf{e}^{(i)})$, *which is convex in* $\mathbf{w}$*. W.r.t. any* $\mathbf{b}_v$*, this has minimizing weights* $\mathbf{w}_v^* := \mathbf{w}_v^*(\mathbf{E}, \mathbf{B}) := \underset{\mathbf{w} \in \mathbb{R}^H}{\arg\min} \ \gamma^{\mathbf{E}}(\mathbf{w}, \mathbf{b}_v)$.

*Then the minimax value of the game* (3) *is* $\mathcal{L}_{\mathbf{B}}^*(\mathbf{E}) = \frac{1}{2}\sum_{v=1}^{V} \gamma^{\mathbf{E}}(\mathbf{w}_v^*, \mathbf{b}_v)$*. For any example* $i \in [n]$*,*

*the minimax optimal reconstruction can be written for any bit* $v$ *as* $\tilde{x}_v^{(i)*} := \frac{1 - e^{-\mathbf{w}_v^{*\top}\mathbf{e}^{(i)}}}{1 + e^{-\mathbf{w}_v^{*\top}\mathbf{e}^{(i)}}}$*.*

This tells us that the optimization problem of finding the minimax optimal reconstructions $\tilde{\mathbf{x}}^{(i)}$ is extremely convenient in several respects. The learning problem decomposes over the $V$ bits in the decoding, reducing to solving for a weight vector $\mathbf{w}_v^* \in \mathbb{R}^H$ for each bit $v$, by optimizing each bitwise slack function. Given the weights, the optimal reconstruction of any example $i$ can be specified by a layer of logistic sigmoid artificial neurons of its encoded bits, with $\mathbf{w}_v^{*\top}\mathbf{e}^{(i)}$ as the bitwise logits.

Hereafter, we write $\mathbf{W} \in \mathbb{R}^{V \times H}$ as the matrix of decoding weights, with rows $\{\mathbf{w}_v\}_{v=1}^V$. In particular, the optimal decoding weights $\mathbf{W}^*(\mathbf{E}, \mathbf{B})$ are the matrix with rows $\{\mathbf{w}_v^*(\mathbf{E}, \mathbf{B})\}_{v=1}^V$.

## 3 LEARNING AN AUTOENCODER

### 3.1 FINDING AN ENCODED REPRESENTATION

Having computed the optimal decoding function in the previous section given any $\mathbf{E}$ and $\mathbf{B}$, we now switch perspectives to the encoder, which seeks to compress the input data $\hat{\mathbf{X}}$ into encoded representations $\mathbf{E}$ (from which $\mathbf{B}$ is easily calculated to pass to the decoder). We seek to find $(\mathbf{E}, \mathbf{B})$ to ensure the lowest worst-case reconstruction loss after decoding; recall that this is $\mathcal{L}_{\mathbf{B}}^*(\mathbf{E})$ from (3).

Observe that $\frac{1}{n}\hat{\mathbf{X}}\mathbf{E}^\top = \mathbf{B}$ by definition, and that the encoder is given $\hat{\mathbf{X}}$. Therefore, by using Thm. 1 and substituting $\mathbf{b}_v = \frac{1}{n}\mathbf{E}\hat{\mathbf{x}}_v \ \forall v \in [V]$,

$$\mathcal{L}_{\mathbf{B}}^*(\mathbf{E}) = \frac{1}{2n}\sum_{i=1}^{n}\sum_{v=1}^{V}\left[-\hat{x}_v^{(i)}(\mathbf{w}_v^{*\top}\mathbf{e}^{(i)}) + \Psi(\mathbf{w}_v^{*\top}\mathbf{e}^{(i)})\right] := \mathcal{L}(\mathbf{W}^*, \mathbf{E}) \tag{4}$$

So it is convenient to define the *feature distortion* [1] for any $v \in [V]$ with respect to $\mathbf{W}$, between any example $\mathbf{x}$ and its encoding $\mathbf{e}$:

$$\beta_v^{\mathbf{W}}(\mathbf{e}, \mathbf{x}) := -x_v \mathbf{w}_v^\top \mathbf{e} + \Psi(\mathbf{w}_v^\top \mathbf{e}) \tag{5}$$

From the above discussion, the best $\mathbf{E}$ given any decoding $\mathbf{W}$, written as $\mathbf{E}^*(\mathbf{W})$, solves the minimization

$$\min_{\mathbf{E} \in [-1,1]^{H \times n}} \mathcal{L}(\mathbf{W}, \mathbf{E}) = \frac{1}{2n} \sum_{i=1}^{n} \min_{\mathbf{e}^{(i)} \in [-1,1]^H} \sum_{v=1}^{V} \beta_v^{\mathbf{W}}(\mathbf{e}^{(i)}, \hat{\mathbf{x}}^{(i)})$$

which immediately yields the following result.

**Proposition 2.** *Define the optimal encodings for decoding weights $\mathbf{W}$ as $\mathbf{E}^*(\mathbf{W}) :=$ $\underset{\mathbf{E} \in [-1,1]^{H \times n}}{\arg\min} \mathcal{L}(\mathbf{W}, \mathbf{E})$. Then $\mathbf{e}^{(i)*}(\mathbf{W})$ can be computed separately for each example $\hat{\mathbf{x}}^{(i)} \in [-1,1]^V$, minimizing its total feature distortion over the decoded bits w.r.t. $\mathbf{W}$:*

$$\text{ENC}(\hat{\mathbf{x}}^{(i)}; \mathbf{W}) := \mathbf{e}^{(i)*}(\mathbf{W}) := \underset{\mathbf{e} \in [-1,1]^H}{\arg\min} \sum_{v=1}^{V} \beta_v^{\mathbf{W}}(\mathbf{e}, \hat{\mathbf{x}}^{(i)}) \tag{6}$$

Observe that the encoding function $\text{ENC}(\hat{\mathbf{x}}^{(i)}; \mathbf{W})$ can be efficiently computed to any desired precision since the feature distortion $\beta_v^{\mathbf{W}}(\mathbf{e}, \hat{\mathbf{x}}^{(i)})$ of each bit $v$ is convex and Lipschitz in $\mathbf{e}$; an $L_1$ error of $\epsilon$ can be reached in $\mathcal{O}(\epsilon^{-2})$ linear-time first-order optimization iterations. Note that the encodings need not be bits, and can be e.g. unconstrained $\in \mathbb{R}^H$ instead; the proof of Thm. 1 assumes no structure on them, and the optimization will proceed as above but without projecting into the hypercube.

## 3.2 AN AUTOENCODER LEARNING ALGORITHM

Our ultimate goal is to minimize the worst-case reconstruction loss. As we have seen in (3) and (6), it is convex in the encoding $\mathbf{E}$ and in the decoding parameters $\mathbf{W}$, each of which can be fixed while minimizing with respect to the other. This suggests a learning algorithm that alternately performs two steps: finding encodings $\mathbf{E}$ that minimize $\mathcal{L}(\mathbf{W}, \mathbf{E})$ as in (6) with a fixed $\mathbf{W}$, and finding decoding parameters $\mathbf{W}^*(\mathbf{E}, \mathbf{B})$, as given in Algorithm 1.

---

**Algorithm 1** Pairwise Correlation Autoencoder (PC-AE)

---

**Input:** Size-$n$ dataset $\hat{\mathbf{X}}$, number of epochs $T$
Initialize $\mathbf{W}_0$ (e.g. with each element being i.i.d. $\sim \mathcal{N}(0,1)$)
**for** $t = 1$ **to** $T$ **do**
 Encode each example to ensure accurate reconstruction using weights $\mathbf{W}_{t-1}$, and compute the associated pairwise bit correlations $\mathbf{B}_t$:

$$\forall i \in [n] : [\mathbf{e}^{(i)}]_t = \text{ENC}(\hat{\mathbf{x}}^{(i)}; \mathbf{W}_{t-1}) \qquad , \qquad \mathbf{B}_t = \frac{1}{n} \hat{\mathbf{X}} \mathbf{E}_t^\top$$

 Update weight vectors $[\mathbf{w}_v]_t$ for each $v \in [V]$ to minimize slack function, using encodings $\mathbf{E}_t$:

$$\forall v \in [V] : [\mathbf{w}_v]_t = \underset{\mathbf{w} \in \mathbb{R}^H}{\arg\min} \left[ -[\mathbf{b}_v]_t^\top \mathbf{w} + \frac{1}{n} \sum_{i=1}^{n} \Psi(\mathbf{w}^\top \mathbf{e}_t^{(i)}) \right]$$

**end for**
**Output:** Weights $\mathbf{W}_T$

---

[1] Noting that $\Psi(\mathbf{w}_v^\top \mathbf{e}) \approx |\mathbf{w}_v^\top \mathbf{e}|$, we see that $\beta_v^{\mathbf{W}}(\mathbf{e}, \hat{\mathbf{x}}) \approx \mathbf{w}_v^\top \mathbf{e} \left( \text{sgn}(\mathbf{w}_v^\top \mathbf{e}) - \hat{x}_v \right)$. So the optimizer tends to change $\mathbf{e}$ so that $\mathbf{w}_v^\top \mathbf{e}$ matches signs with $\hat{x}_v$, motivating the name.

### 3.3 Efficient Implementation

Our derivation of the encoding and decoding functions involves no model assumptions at all, only using the minimax structure and pairwise statistics that the algorithm is allowed to remember. Nevertheless, the (en/de)coders can be learned and implemented efficiently.

Decoding is a convex optimization in $H$ dimensions, which can be done in parallel for each bit $v \in [V]$. This is relatively easy to solve in the parameter regime of primary interest when data are abundant, in which $H < V \ll n$. Similarly, encoding is also a convex optimization problem in only $H$ dimensions. If the data examples are instead sampled in minibatches of size $n$, they can be encoded in parallel, with a new minibatch being sampled to start each epoch $t$. The number of examples $n$ (per batch) is essentially only limited by $nH$, the number of compressed representations that fit in memory.

So far in this paper, we have stated our results in the transductive setting, in which all data are given together a priori, with no assumptions whatsoever made about the interdependences between the $V$ features. However, PC-AE operates much more efficiently than this might suggest. Crucially, the encoding and decoding tasks both depend on $n$ only to average a function of $\mathbf{x}^{(i)}$ or $\mathbf{e}^{(i)}$ over $i \in [n]$, so they can both be solved by stochastic optimization methods that use first-order gradient information, like variants of stochastic gradient descent (SGD). We find it remarkable that the minimax optimal encoding and decoding can be efficiently learned by such methods, which do not scale computationally in $n$. Note that the result of each of these steps involves $\Omega(n)$ outputs ($\mathbf{E}$ and $\tilde{\mathbf{X}}$), which are all coupled together in complex ways.

Furthermore, efficient first-order convex optimization methods for both encoding and decoding steps manipulate more intermediate gradient-related quantities, with facile interpretations. For details, see Appendix A.2.

### 3.4 Convergence and Weight Regularization

As we noted previously, the objective function of the optimization is biconvex. This means that the alternating minimization algorithm we specify is an instance of *alternating convex search*, shown in that literature to converge under broad conditions (Gorski et al. (2007)). It is not guaranteed to converge to the global optimum, but each iteration will monotonically decrease the objective function. In light of our introductory discussion, the properties and rate of such convergence would be interesting to compare to stochastic optimization algorithms for PCA, which converge efficiently under broad conditions (Balsubramani et al. (2013); Shamir (2016)).

The basic game used so far has assumed perfect knowledge of the pairwise correlations, leading to equality constraints $\forall v \in [V]: \quad \frac{1}{n}\mathbf{E}\mathbf{x}_v = \mathbf{b}_v$. This makes sense in PC-AE, where the encoding phase of each epoch gives the exact $\mathbf{B}_t$ for the decoding phase. However, in other stochastic settings as for denoising autoencoders (see Sec. 5.2), it may be necessary to relax this constraint. A relaxed constraint of $\left\| \frac{1}{n}\mathbf{E}\mathbf{x}_v - \mathbf{b}_v \right\|_\infty \leq \epsilon$ exactly corresponds to an extra additive regularization term of $\epsilon \left\| \mathbf{w}_v \right\|_1$ on the corresponding weights in the convex optimization used to find $\mathbf{W}$ (Appendix D.1). Such regularization leads to provably better generalization (Bartlett (1998)) and is often practical to use, e.g. to encourage sparsity. But we do not use it for our PC-AE experiments in this paper.

## 4 Discussion and Related Work

Our approach PC-AE is quite different from existing autoencoding work in several ways.

First and foremost, we posit no explicit decision rule, and avoid optimizing the highly non-convex decision surface traversed by traditional autoencoding algorithms that learn with backpropagation (Rumelhart et al. (1986)). The decoding function, given the encodings, is a single layer of artificial neurons only because of the minimax structure of the problem when minimizing worst-case loss. This differs from reasoning typically used in neural net work (see Jordan (1995)), in which the loss is the negative log-likelihood (NLL) of the joint probability, which is *assumed* to follow a form specified by logistic artificial neurons and their weights. We instead interpret the loss in the usual direct way as the NLL of the predicted probability of the data given the visible bits, and avoid any assumptions on the decision rule (e.g. not monotonicity in the score $\mathbf{w}_v^\top \mathbf{e}^{(i)}$, or even dependence on such a score).

This justification of artificial neurons – as the minimax optimal decision rules given information on pairwise correlations – is one of our more distinctive contributions (see Sec. 5.1).

Crucially, we make no assumptions whatsoever on the form of the encoding or decoding, except on the memory used by the decoding. Some such "regularizing" restriction is necessary to rule out the autoencoder just memorizing the data, and is typically expressed by assuming a model class of compositions of artificial neuron layers. We instead impose it axiomatically by limiting the amount of information transmitted through $\mathbf{B}$, which does not scale in $n$; but we do not restrict how this information is used. This confers a clear theoretical advantage, allowing us to attain the strongest robust loss guarantee among *all possible* autoencoders that use the correlations $\mathbf{B}$.

More importantly in practice, avoiding an explicit model class means that we do not have to optimize the typically non-convex model, which has long been a central issue for backpropagation-based learning methods (e.g. Dauphin et al. (2014)). Prior work related in spirit has attempted to avoid this through convex relaxations, including for multi-layer optimization under various structural assumptions (Aslan et al. (2014); Zhang et al. (2016)), and when the number of hidden units is varied by the algorithm (Bengio et al. (2005); Bach (2014)).

Our approach also isolates the benefit of higher $n$ in dealing with overfitting, as the pairwise correlations $\mathbf{B}$ can be measured progressively more accurately as $n$ increases. In this respect, we follow a line of research using such pairwise correlations to model arbitary higher-order structure among visible units, rooted in early work on (restricted) Boltzmann Machines (Ackley et al. (1985); Smolensky (1986); Rumelhart & McClelland (1987); Freund & Haussler (1992)). More recently, theoretical algorithms have been developed with the perspective of learning from the correlations between units in a network, under various assumptions on the activation function, architecture, and weights, for both deep (Arora et al. (2014)) and shallow networks (using tensor decompositions, e.g. Livni et al. (2014); Janzamin et al. (2015)). Our use of ensemble aggregation techniques (from Balsubramani & Freund (2015a; 2016)) to study these problems is anticipated in spirit by prior work as well, as discussed at length by Bengio (2009) in the context of distributed representations.

## 4.1 OPTIMALITY, OTHER ARCHITECTURES, AND DEPTH

We have established that a single layer of logistic artificial neurons is an optimal decoder, given only indirect information about the data through pairwise correlations. This is not a claim that autoencoders need only a single-layer architecture in the worst case. Sec. 3.1 establishes that the best representations $\mathbf{E}$ are the solution to a convex optimization, with no artificial neurons involved in computing them from the data. Unlike the decoding function, the optimal encoding function ENC cannot be written explicitly in terms of artificial neurons, and is incomparable to existing architectures (though it is analogous to PCA in prescribing an efficient operation that yields the encodings from unlabeled data). Also, the encodings are only optimal given the pairwise correlations; training algorithms like backpropagation, which communicate other knowledge of the data through derivative composition, can learn final decoding layers that outperform ours, as we see in experiments.

In our framework so far, we explore using all the pairwise correlations between hidden and visible bits to inform learning by constraining the adversary, resulting in a Lagrange parameter – a weight – for each constraint. These $VH$ weights $\mathbf{W}$ constitute the parameters of the optimal decoding layer, describing a fully connected architecture. If just a select few of these correlations were used, only they would constrain the adversary in the minimax problem of Sec. 2, so weights would only be introduced for them, giving rise to sparser architectures.

Our central choices – to store only pairwise correlations and minimize worst-case reconstruction loss – play a similar regularizing role to explicit model assumptions, and other autoencoding methods may achieve better performance on data for which these choices are too conservative, by e.g. making distributional assumptions on the data. From our perspective, other architectures with more layers – particularly highly successful ones like convolutional, recurrent, residual, and ladder networks (LeCun et al. (2015); He et al. (2015); Rasmus et al. (2015)) – lend the autoencoding algorithm more power by allowing it to measure more nuanced correlations using more parameters, which decreases the worst-case loss. Applying our approach with these would be interesting future work.

Extending this paper's convenient minimax characterization to deep representations with empirical success is a very interesting open problem. Prior work on stacking autoencoders/RBMs (Vincent et al.

(2010)) and our learning algorithm PC-AE suggest that we could train a deep network in alternating forward and backward passes. Using this paper's ideas, the forward pass would learn the weights to each layer given the previous layer's activations (and inter-layer pairwise correlations) by minimizing the slack function, with the backward pass learning the activations for each layer given the weights to / activations of the next layer by convex optimization (as we learn $\mathbf{E}$). Both passes would consist of successive convex optimizations dictated by our approach, quite distinct from backpropagation, though loosely resembling the wake-sleep algorithm (Hinton et al. (1995)).

## 4.2 GENERATIVE APPLICATIONS

Particularly recently, autoencoders have been of interest largely for their many applications beyond compression, especially for their generative uses. The most directly relevant to us involve repurposing denoising autoencoders (Bengio et al. (2013b); see Sec. 5.2); moment matching among hidden and visible units (Li et al. (2015)); and generative adversarial network ideas (Goodfellow et al. (2014); Makhzani et al. (2015)), the latter particularly since the techniques of this paper have been applied to binary classification (Balsubramani & Freund (2015a;b)). These are outside this paper's scope, but suggest themselves as future extensions of our approach.

## 5 EXTENSIONS

### 5.1 OTHER RECONSTRUCTION LOSSES

It may make sense to use another reconstruction loss other than cross-entropy, for instance the expected Hamming distance between $\mathbf{x}^{(i)}$ and $\tilde{\mathbf{x}}^{(i)}$. It turns out that the minimax manipulations we use work under very broad conditions, for nearly any loss that additively decomposes over the $V$ bits as cross-entropy does. In such cases, all that is required is that the partial losses $\ell_+(\tilde{x}_v^{(i)}), \ell_-(\tilde{x}_v^{(i)})$ are monotonically decreasing and increasing respectively (recall that for cross-entropy loss, this is true as $\ell_\pm(\tilde{x}_v^{(i)}) = \ln\left(\frac{2}{1\pm\tilde{x}_v^{(i)}}\right)$); they need not even be convex. This monotonicity is a natural condition, because the loss measures the discrepancy to the true label, and holds for all losses in common use.

Changing the partial losses only changes the structure of the minimax solution in two respects: by altering the form of the transfer function on the decoding neurons, and the univariate potential well $\Psi$ optimized to learn the decoding weights. Otherwise, the problem remains convex and the algorithm is identical. Formal statements of these general results are in Appendix E.

### 5.2 DENOISING AUTOENCODING

Our framework can be easily applied to learn a denoising autoencoder (DAE; Vincent et al. (2008; 2010)), which uses noise-corrupted data (call it $\dot{\mathbf{X}}$) for training, and uncorrupted data for evaluation. From our perspective, this corresponds to leaving the learning of $\mathbf{W}$ unchanged, but using corrupted data when learning $\mathbf{E}$. Consequently, the minimization problem over encodings must be changed to account for the bias on $\mathbf{B}$ introduced by the noise; so the algorithm plays given the noisy data, but to minimize loss against $\mathbf{X}$. This is easiest to see for zero-mean noise, for which our algorithms are completely unchanged because $\mathbf{B}$ does not change (in expectation) after the noise is added.

Another common scenario illustrating this technique is to mask a $\rho$ fraction of the input bits uniformly at random (in our notation, changing 1s to $-1$s). This masking noise changes each pairwise correlation $b_{v,h}$ by an amount $\delta_{v,h} := \frac{1}{n}\sum_{i=1}^{n}(\dot{x}_v^{(i)} - x_v^{(i)})e_h^{(i)}$. Therefore, the optimand Eq. (4) must be modified by subtracting this factor $\delta_{v,h}$. This $\delta_{v,h}$ can be estimated (w.h.p.) given $\dot{\mathbf{x}}_v, \mathbf{e}_h, \rho, \mathbf{x}_v$. But even with just the noisy data and not $\mathbf{x}_v$, we can estimate $\delta_{v,h}$ w.h.p. by extrapolating the correlation of the bits of $\dot{\mathbf{x}}_v$ that are left as $+1$ (a $1-\rho$ fraction) with the corresponding values in $\mathbf{e}_h$ (see Appendix C).

Table 1: Cross-entropy reconstruction losses for PC-AE and a vanilla single-layer autoencoder, with binary and unconstrained real-valued encodings, and significant results in bold. The PC-AE results are significantly better (see Appendix A) than the AE results.

|  | PC-AE (bin.) | PC-AE (real) | AE (bin.) | AE (real) | PCA |
|---|---|---|---|---|---|
| MNIST, $H = 32$ | **51.9** | **53.8** | 65.2 | 64.3 | 86.6 |
| MNIST, $H = 100$ | **9.2** | **9.9** | 26.8 | 25.0 | 52.7 |
| Omniglot, $H = 32$ | **76.1** | **77.2** | 93.1 | 90.6 | 102.8 |
| Omniglot, $H = 100$ | **12.1** | **13.2** | 46.6 | 45.4 | 63.6 |
| Caltech-101, $H = 32$ | **54.5** | **54.9** | 97.5 | 87.6 | 118.7 |
| Caltech-101, $H = 100$ | **7.1** | **7.1** | 64.3 | 45.4 | 75.2 |
| notMNIST, $H = 32$ | **121.9** | **122.4** | 149.6 | 141.8 | 174.0 |
| notMNIST, $H = 100$ | **62.2** | **63.0** | 99.6 | 92.1 | 115.5 |
| Adult, $H = 10$ | **7.7** | **7.8** | **9.3** | **8.1** | 13.5 |
| Adult, $H = 20$ | **0.65** | **0.64** | 2.5 | **1.5** | 7.9 |

# 6 EXPERIMENTS

In this section we compare our approach [2] empirically to a standard autoencoder with one hidden layer (termed AE here) trained with backpropagation, and a thresholded PCA baseline. Our goal is simply to verify that our approach, though very different, is competitive in reconstruction performance.

The datasets we use are first normalized to $[0, 1]$, and then binarized by sampling each pixel stochastically in proportion to its intensity, following prior work (Salakhutdinov & Murray (2008)). Changing between binary and real-valued encodings in PC-AE requires just a line of code, to project the encodings into $[-1, 1]^H$ after convex optimization updates to compute ENC($\cdot$). We use Adagrad (Duchi et al. (2011)) for the convex minimizations of our algorithms; we observed that their performance is not very sensitive to the choice of optimization method, explained by our approach's convexity.

We compare to a basic AE with a single hidden layer, trained using the Adam method with default parameters (Kingma & Ba (2014)). Other models like variational autoencoders (Kingma & Welling (2013)) are not shown here because they do not aim to optimize reconstruction loss or are not comparably general autoencoding architectures. We also use a sign-thresholded PCA baseline (essentially a completely linear autoencoder, but with the output layer thresholded to be in $[-1, 1]$); see Appendix A for more details. We vary the number of hidden units $H$ for all algorithms, and try both binary and unconstrained real-valued encodings where appropriate; the respective AE uses logistic sigmoid and ReLU transfer functions for the encoding neurons. The results are in Table 1.

The reconstruction performance of PC-AE indicates that it can encode information very well using pairwise correlations, compared to the directly learned AE and PCA approaches. Loss can become extremely low when $H$ is raised, giving $\mathbf{B}$ the capacity to robustly encode almost all the information in the input bits $\hat{\mathbf{X}}$. The performance is roughly equal between binary hidden units and unconstrained ones, which is expected by our derivations.

We also try learning just the decoding layer of Sec. 2, on the encoded representation of the AE. This is motivated by the fact that Sec. 2 establishes our decoding method to be worst-case optimal given any $\mathbf{E}$ and $\mathbf{B}$. We find the results to be significantly worse than the AE alone in all datasets used (e.g. reconstruction loss of $\sim 171/133$ on MNIST, and $\sim 211/134$ on Omniglot, with 32/100 hidden units respectively). This reflects the AE's training backpropagating information about the data beyond pairwise correlations, through non-convex function compositions – however, this comes at the cost of being more difficult to optimize. The representations learned by the ENC function of PC-AE are quite different and capture much more of the pairwise correlation information, which is used by the decoding layer in a worst-case optimal fashion. We attempt to visually depict the differences between the representations in Fig. 3.

As discussed in Sec. 4, we do not claim that this paper's method will always achieve the best empirical reconstruction loss, even among single-layer autoencoders. We would like to make the encoding

---

[2]TensorFlow code available at `https://github.com/aikanor/pc-autoencoder`.

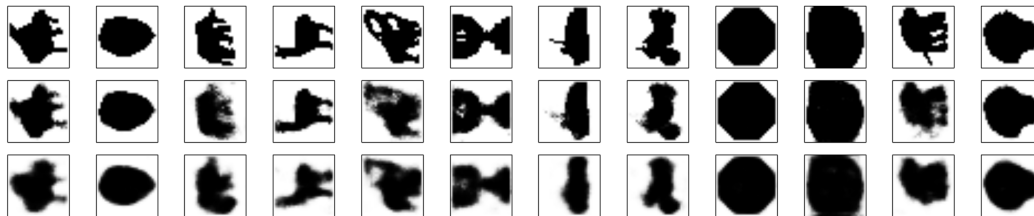

Figure 1: Top row: randomly chosen test images from Caltech-101 silhouettes. Middle and bottom rows: corresponding reconstructions of PC-AE and AE with $H = 32$ binary hidden units.

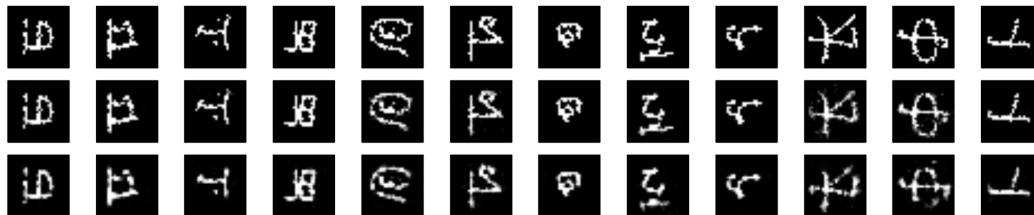

Figure 2: As Fig. 2, with $H = 100$ on Omniglot. Difference in quality is particularly noticeable in the 1st, 5th, 8th, and 11th columns.

function quicker to compute, as well. But we believe this paper's results, especially when $H$ is high, illustrate the potential of using pairwise correlations for autoencoding as in our approach, learning to encode with alternating convex minimization and extremely strong worst-case robustness guarantees.

ACKNOWLEDGMENTS

I am grateful to Jack Berkowitz, Sanjoy Dasgupta, and Yoav Freund for helpful discussions; Daniel Hsu and Akshay Krishnamurthy for instructive examples; and Gary Cottrell for an enjoyable chat. I acknowledge funding from the NIH (grant R01ES02500902).

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

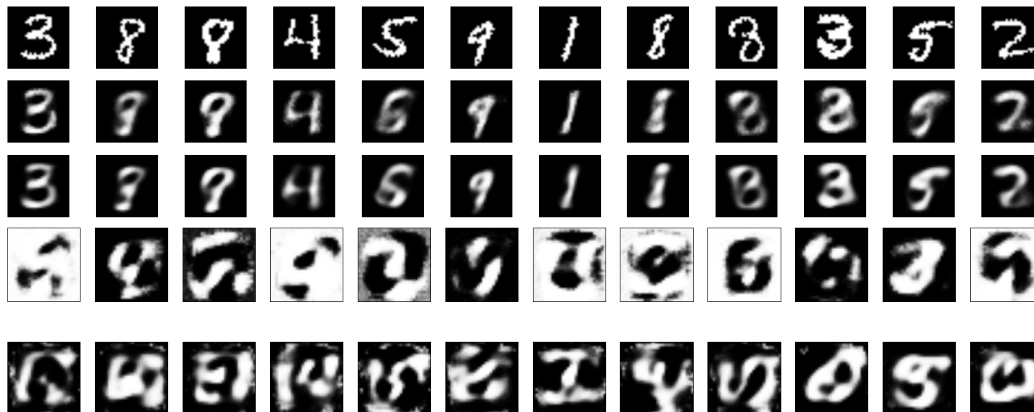

Figure 3: Top three rows: the reconstructions of random test images from MNIST ($H = 12$), as in Fig. 2. PC-AE achieves loss $105.1$ here, and AE $111.2$. Fourth and fifth rows: visualizations of all the hidden units of PC-AE and AE, respectively. It is not possible to visualize the PC-AE encoding units by the image that maximally activates them, as commonly done, because of the form of the ENC function which depends on $\mathbf{W}$ and lacks explicit encoding weights. So each hidden unit $h$ is depicted by the visible decoding of the encoded representation which has bit $h$ "on" and all other bits "off." (If this were PCA with a linear decoding layer, this would simply represent hidden unit $h$ by its corresponding principal component vector, the decoding of the $h^{th}$ canonical basis vector in $\mathbb{R}^H$.)

Akshay Balsubramani and Yoav Freund. Optimal binary classifier aggregation for general losses. In *Advances in Neural Information Processing Systems (NIPS)*, 2016. arXiv:1510.00452.

Akshay Balsubramani, Sanjoy Dasgupta, and Yoav Freund. The fast convergence of incremental pca. In *Advances in Neural Information Processing Systems (NIPS)*, pp. 3174–3182, 2013.

Peter L Bartlett. The sample complexity of pattern classification with neural networks: the size of the weights is more important than the size of the network. *IEEE Transactions on Information Theory*, 44(2):525–536, 1998.

Yoshua Bengio. Learning deep architectures for ai. *Foundations and Trends in Machine Learning*, 2 (1):1–127, 2009.

Yoshua Bengio, Nicolas L Roux, Pascal Vincent, Olivier Delalleau, and Patrice Marcotte. Convex neural networks. In *Advances in neural information processing systems (NIPS)*, pp. 123–130, 2005.

Yoshua Bengio, Aaron Courville, and Pierre Vincent. Representation learning: A review and new perspectives. *Pattern Analysis and Machine Intelligence, IEEE Transactions on*, 35(8):1798–1828, 2013a.

Yoshua Bengio, Li Yao, Guillaume Alain, and Pascal Vincent. Generalized denoising auto-encoders as generative models. In *Advances in Neural Information Processing Systems (NIPS)*, pp. 899–907, 2013b.

Hervé Bourlard and Yves Kamp. Auto-association by multilayer perceptrons and singular value decomposition. *Biological cybernetics*, 59(4-5):291–294, 1988.

Yuri Burda, Roger Grosse, and Ruslan Salakhutdinov. Importance weighted autoencoders. *International Conference on Learning Representations (ICLR)*, 2016. arXiv preprint arXiv:1509.00519.

Nicolo Cesa-Bianchi and Gàbor Lugosi. *Prediction, Learning, and Games*. Cambridge University Press, New York, NY, USA, 2006.

Yann N Dauphin, Razvan Pascanu, Caglar Gulcehre, Kyunghyun Cho, Surya Ganguli, and Yoshua Bengio. Identifying and attacking the saddle point problem in high-dimensional non-convex optimization. In *Advances in neural information processing systems (NIPS)*, pp. 2933–2941, 2014.

John Duchi, Elad Hazan, and Yoram Singer. Adaptive subgradient methods for online learning and stochastic optimization. *The Journal of Machine Learning Research*, 12:2121–2159, 2011.

Yoav Freund and David Haussler. Unsupervised learning of distributions on binary vectors using two layer networks. In *Advances in Neural Information Processing Systems (NIPS)*, pp. 912–919, 1992.

Ian Goodfellow, Jean Pouget-Abadie, Mehdi Mirza, Bing Xu, David Warde-Farley, Sherjil Ozair, Aaron Courville, and Yoshua Bengio. Generative adversarial nets. In *Advances in Neural Information Processing Systems (NIPS)*, pp. 2672–2680, 2014.

Jochen Gorski, Frank Pfeuffer, and Kathrin Klamroth. Biconvex sets and optimization with biconvex functions: a survey and extensions. *Mathematical Methods of Operations Research*, 66(3):373–407, 2007.

Kaiming He, Xiangyu Zhang, Shaoqing Ren, and Jian Sun. Deep residual learning for image recognition. *arXiv preprint arXiv:1512.03385*, 2015.

Geoffrey E Hinton, Peter Dayan, Brendan J Frey, and Radford M Neal. The" wake-sleep" algorithm for unsupervised neural networks. *Science*, 268(5214):1158–1161, 1995.

Majid Janzamin, Hanie Sedghi, and Anima Anandkumar. Beating the perils of non-convexity: Guaranteed training of neural networks using tensor methods. *arXiv preprint arXiv:1506.08473*, 2015.

Michael I Jordan. Why the logistic function? a tutorial discussion on probabilities and neural networks, 1995.

Diederik Kingma and Jimmy Ba. Adam: A method for stochastic optimization. *arXiv preprint arXiv:1412.6980*, 2014.

Diederik P Kingma and Max Welling. Auto-encoding variational bayes. *arXiv preprint arXiv:1312.6114*, 2013.

Yann LeCun, Yoshua Bengio, and Geoffrey Hinton. Deep learning. *Nature*, 521(7553):436–444, 2015.

Yujia Li, Kevin Swersky, and Rich Zemel. Generative moment matching networks. In *Proceedings of the 32nd International Conference on Machine Learning (ICML-15)*, pp. 1718–1727, 2015.

Roi Livni, Shai Shalev-Shwartz, and Ohad Shamir. On the computational efficiency of training neural networks. In *Advances in Neural Information Processing Systems (NIPS)*, pp. 855–863, 2014.

Alireza Makhzani, Jonathon Shlens, Navdeep Jaitly, and Ian Goodfellow. Adversarial autoencoders. *arXiv preprint arXiv:1511.05644*, 2015.

Antti Rasmus, Mathias Berglund, Mikko Honkala, Harri Valpola, and Tapani Raiko. Semi-supervised learning with ladder networks. In *Advances in Neural Information Processing Systems*, pp. 3546–3554, 2015.

David E Rumelhart and James L McClelland. Parallel distributed processing, explorations in the microstructure of cognition. vol. 1: Foundations. *Computational Models of Cognition and Perception, Cambridge: MIT Press*, 1987.

David E Rumelhart, Geoffrey E Hinton, and Ronald J Williams. Learning representations by back-propagating errors. *Nature*, 323(6088):533–536, 1986.

Ruslan Salakhutdinov and Iain Murray. On the quantitative analysis of deep belief networks. In *Proceedings of the 25th International Conference on Machine Learning (ICML)*, pp. 872–879, 2008.

Ohad Shamir. Convergence of stochastic gradient descent for pca. *International Conference on Machine Learning (ICML)*, 2016. arXiv preprint arXiv:1509.09002.

P Smolensky. Information processing in dynamical systems: foundations of harmony theory. In *Parallel distributed processing: explorations in the microstructure of cognition, vol. 1*, pp. 194–281. MIT Press, 1986.

Pascal Vincent, Hugo Larochelle, Yoshua Bengio, and Pierre-Antoine Manzagol. Extracting and composing robust features with denoising autoencoders. In *Proceedings of the 25th international conference on Machine learning (ICML)*, pp. 1096–1103. ACM, 2008.

Pascal Vincent, Hugo Larochelle, Isabelle Lajoie, Yoshua Bengio, and Pierre-Antoine Manzagol. Stacked denoising autoencoders: Learning useful representations in a deep network with a local denoising criterion. *The Journal of Machine Learning Research*, 11:3371–3408, 2010.

Yuchen Zhang, Percy Liang, and Martin J Wainwright. Convexified convolutional neural networks. *arXiv preprint arXiv:1609.01000*, 2016.

## A    EXPERIMENTAL DETAILS

In addition to MNIST, we use the preprocessed version of the Omniglot dataset found in Burda et al. (2016), split 1 of the Caltech-101 Silhouettes dataset, the small notMNIST dataset, and the UCI Adult (a1a) dataset. The results reported are the mean of 10 Monte Carlo runs, and the PC-AE significance results use 95% Monte Carlo confidence intervals. Only notMNIST comes without a predefined split, so the displayed results use 10-fold cross-validation. Non-binarized versions of all datasets (grayscale pixels) resulted in nearly identical PC-AE performance (not shown); this is as expected from its derivation using expected pairwise correlations, which with high probability are nearly invariant under binarization (by e.g. Hoeffding bounds).

We used minibatches of size 250. All standard autoencoders use the 'Xavier' initialization and trained for 500 epochs or using early stopping on the test set. The "PCA" baseline was run on exactly the same input data as the others; it finds decodings by mean-centering this input, finding the top $H$ principal components with standard PCA, reconstructing the mean-centered input with these components, adding back the means, and finally thresholding the result to $[-1, 1]^V$.

We did not evaluate against other types of autoencoders which regularize (Kingma & Welling (2013)) or are otherwise not trained for direct reconstruction loss minimization. Also, not shown is the performance of a standard convolutional autoencoder (32-bit representation, depth-3 64-64-32 (en/de)coder) which performs better than the standard autoencoder, but is still outperformed by PC-AE on our image-based datasets. A deeper architecture could quite possibly achieve superior performance, but the greater number of channels through which information is propagated makes fair comparison with our flat fully-connected approach difficult. We consider extension of our PC-AE approach to such architectures to be fascinating future work.

### A.1    FURTHER RESULTS

Our bound on worst-case loss is invariably quite tight, as shown in Fig. 4. Similar results are found on all datasets. This is consistent with our conclusions about the nature of the PC-AE representations – conveying almost exactly the information available in pairwise correlations.

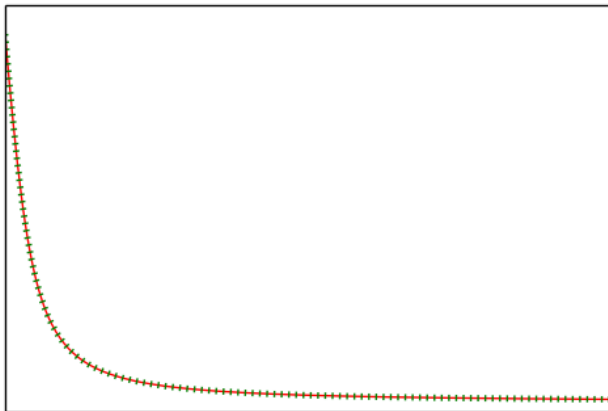

Figure 4: Actual reconstruction loss to real data (red) and slack function [objective function] value (dotted green), during Adagrad optimization to learn $\mathbf{W}$ using the optimal $\mathbf{E}, \mathbf{B}$. Monotonicity is expected since this is a convex optimization. The objective function value theoretically upper-bounds the actual loss, and practically tracks it nearly perfectly.

A 2D visualization of MNIST is in Fig. 6, showing that even with just two hidden units there is enough information in pairwise correlations for PC-AE to learn a sensible embedding. We also include more pictures of our autoencoders' reconstructions, and visualizations of the hidden units when $H = 100$ in Fig. 5.

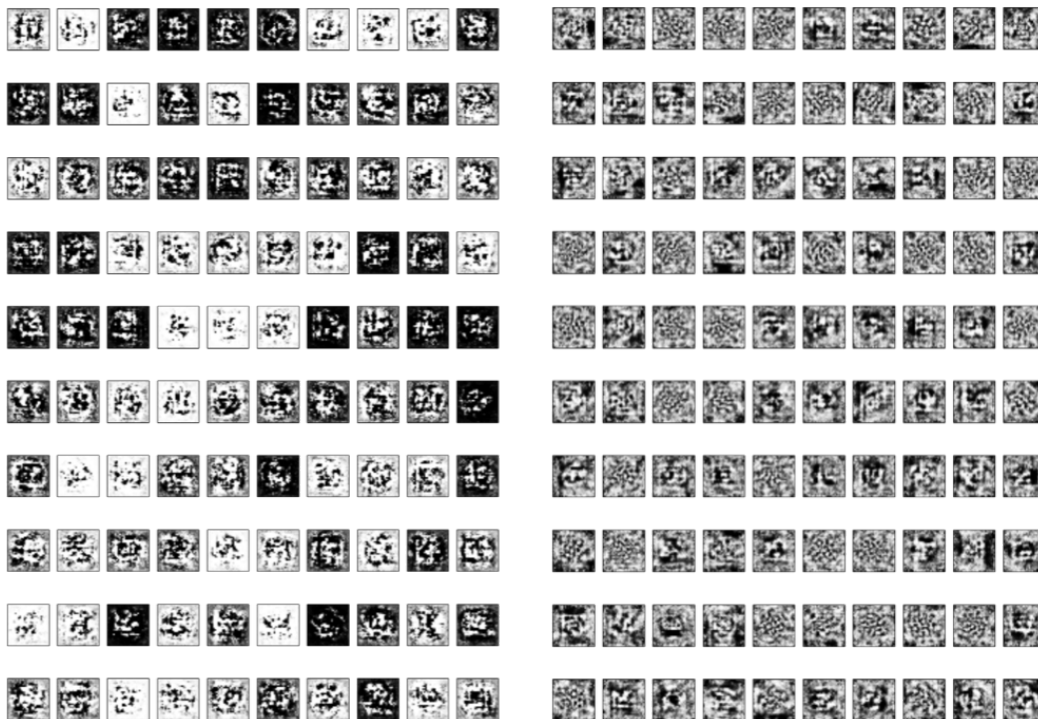

Figure 5: Visualizations of all the hidden units of PC-AE (left) and AE (right) from Omniglot for $H = 100$, as in Fig. 3.

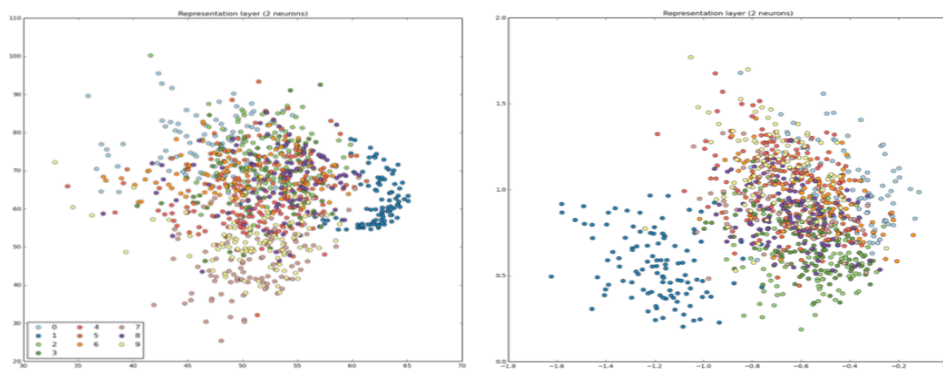

Figure 6: AE (left) and PC-AE (right) visualizations of a random subset of MNIST test data, with $H = 2$ real-valued hidden units, and colors corresponding to class labels (legend at left). PC-AE's loss is $\sim 189$ here, and that of AE is $\sim 179$.

## A.2 PC-AE INTERPRETATION AND IMPLEMENTATION DETAILS

Here we give some details that are useful for interpretation and implementation of the proposed method.

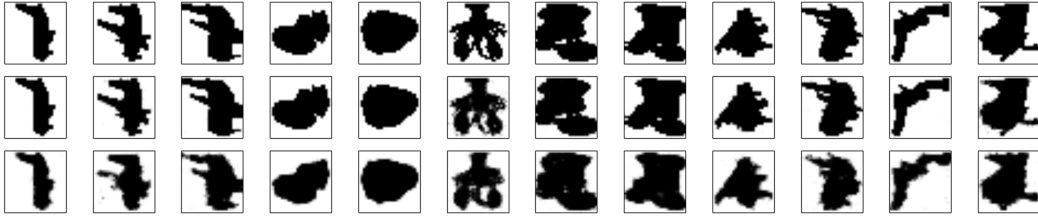

Figure 7: As Fig. 2, with $H = 100$ on Caltech-101 silhouettes.

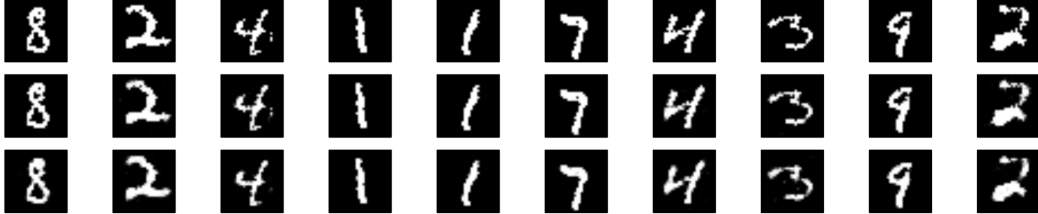

Figure 8: As Fig. 2, with $H = 100$ on MNIST.

### A.2.1 ENCODING

Proposition 2 defines the encoding function for any data example $\mathbf{x}$ as the vector that minimizes the total feature distortion, summed over the bits in the decoding, rewritten here for convenience:

$$\text{ENC}(\mathbf{x}^{(i)}; \mathbf{W}) := \underset{\mathbf{e} \in [-1,1]^H}{\arg\min} \sum_{v=1}^{V} \left[ -x_v^{(i)} \mathbf{w}_v^\top \mathbf{e}^{(i)} + \Psi(\mathbf{w}_v^\top \mathbf{e}^{(i)}) \right] \tag{7}$$

Doing this on multiple examples at once (in memory as a minibatch) can be much faster than on each example separately. We can now compute the gradient of the objective function w.r.t. each example $i \in [n]$, writing the gradient w.r.t. example $i$ as column $i$ of a matrix $\mathbf{G} \in \mathbb{R}^{H \times n}$. $\mathbf{G}$ can be calculated efficiently in a number of ways, for example as follows:

- Compute matrix of **hallucinated data** $\breve{\mathbf{X}} := \Psi'(\mathbf{W}\mathbf{E}) \in \mathbb{R}^{V \times n}$.
- Subtract $\mathbf{X}$ to compute **residuals** $\mathbf{R} := \breve{\mathbf{X}} - \mathbf{X} \in \mathbb{R}^{V \times n}$.
- Compute $\mathbf{G} = \frac{1}{n} \mathbf{W}^\top \mathbf{R} \in \mathbb{R}^{H \times n}$.

Optimization then proceeds with gradient descent using $\mathbf{G}$, with the step size found using line search. Note that since the objective function is convex, the optimum $\mathbf{E}^*$ leads to optimal residuals $\mathbf{R}^* \in \mathbb{R}^{V \times n}$ such that $\mathbf{G} = \frac{1}{n} \mathbf{W}^\top \mathbf{R}^* = \mathbf{0}^{H \times n}$, so each column of $\mathbf{R}^*$ is in the null space of $\mathbf{W}^\top$, which maps the residual vectors to the encoded space. We conclude that although the compression is not perfect (so the optimal residuals $\mathbf{R}^* \neq \mathbf{0}^{V \times n}$ in general), each column of $\mathbf{R}^*$ is orthogonal to the decoding weights at an equilibrium towards which the convex minimization problem of (7) is guaranteed to stably converge.

### A.2.2 DECODING

The decoding step finds $\mathbf{W}$ to ensure accurate decoding of the given encodings $\mathbf{E}$ with correlations $\mathbf{B}$, solving the convex minimization problem:

$$\mathbf{W}^* = \underset{\mathbf{W} \in \mathbb{R}^{V \times H}}{\arg\min} \sum_{v=1}^{V} \left[ -\mathbf{b}_v^\top \mathbf{w}_v + \frac{1}{n} \sum_{i=1}^{n} \Psi(\mathbf{w}_v^\top \mathbf{e}^{(i)}) \right] \tag{8}$$

This can be minimized by first-order convex optimization. The gradient of (8) at $\mathbf{W}$ is:

$$-\mathbf{B} + \frac{1}{n} [\Psi'(\mathbf{W}\mathbf{E})] \mathbf{E}^\top \tag{9}$$

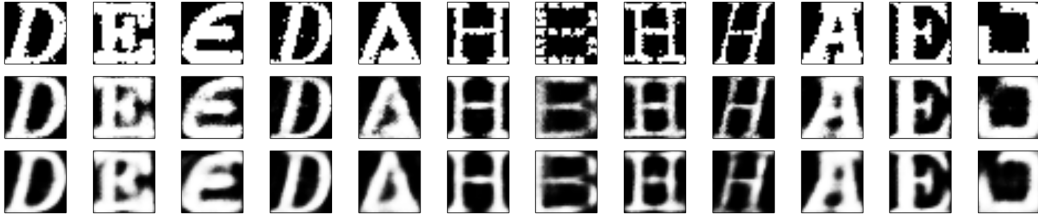

Figure 9: As Fig. 2, with $H = 32$ on notMNIST.

The second term can be understood as "hallucinated" pairwise correlations $\breve{\mathbf{B}}$, between bits of the encoded examples $\mathbf{E}$ and bits of their decodings under the current weights, $\breve{\mathbf{X}} := \Psi'(\mathbf{WE})$. The hallucinated correlations can be written as $\breve{\mathbf{B}} := \frac{1}{n}\breve{\mathbf{X}}\mathbf{E}^\top$. Therefore, (9) can be interpreted as the residual correlations $\breve{\mathbf{B}} - \mathbf{B}$. Since the slack function of (8) is convex, the optimum $\mathbf{W}^*$ leads to hallucinated correlations $\breve{\mathbf{B}}^* = \mathbf{B}$, which is the limit reached by the optimization algorithm after many iterations.

## B  ALLOWING RANDOMIZED DATA AND ENCODINGS

In this paper, we represent the bit-vector data in a randomized way in $[-1, 1]^V$. Randomizing the data only relaxes the constraints on the adversary in the game we play; so at worst we are working with an upper bound on worst-case loss, instead of the exact minimax loss itself, erring on the conservative side. Here we briefly justify the bound as being essentially tight, which we also see empirically in this paper's experiments.

In the formulation of Section 2, the only information we have about the data is its pairwise correlations with the encoding units. When the data are abundant ($n$ large), then w.h.p. these correlations are close to their expected values over the data's internal randomization, so representing them as continuous values w.h.p. results in the same $\mathbf{B}$ and therefore the same solutions for $\mathbf{E}, \mathbf{W}$. We are effectively allowing the adversary to play each bit's conditional probability of firing, rather than the binary realization of that probability.

This allows us to apply minimax theory and duality to considerably simplify the problem to a convex optimization, when it would otherwise be nonconvex, and computationally hard (Baldi (2012)). The fact that we are only using information about the data through its *expected* pairwise correlations with the hidden units makes this possible.

The above also applies to the encodings and their internal randomization, allowing us to learn binary randomized encodings by projecting to the convex set $[-1, 1]^H$.

## C  DENOISING AUTOENCODER WITH MASKING NOISE: DETAILS

This section elaborates on the discussion of Sec. 5.2.

Recall the correlation correction term $\delta_{v,h}$ from Sec. 5.2:

$$\delta_{v,h} = \frac{1}{n}\sum_{i=1}^{n}(\dot{x}_v^{(i)} - x_v^{(i)})e_h^{(i)}$$

Here, we express this in terms of the known quantities $\dot{\mathbf{x}}_v, \mathbf{e}_h, \rho$, and not the unknown denoised data $\mathbf{x}_v$.

Consider that

$$(\dot{x}_v^{(i)} - x_v^{(i)})e_h^{(i)} = \mathbf{1}\left(x_v^{(i)} = -1\right)(\dot{x}_v^{(i)} - x_v^{(i)})e_h^{(i)} + \mathbf{1}\left(x_v^{(i)} = +1\right)(\dot{x}_v^{(i)} - x_v^{(i)})e_h^{(i)}$$

Now if $x_v^{(i)} = -1$, then $\dot{x}_v^{(i)} = -1$, so $(\dot{x}_v^{(i)} - x_v^{(i)})e_h^{(i)} = 0$. Therefore the first term above is zero, and the expression can be simplified:

$$(\dot{x}_v^{(i)} - x_v^{(i)})e_h^{(i)} = \mathbf{1}\left(x_v^{(i)} = +1\right)(\dot{x}_v^{(i)} - x_v^{(i)})e_h^{(i)} = \mathbf{1}\left(x_v^{(i)} = +1 \wedge \dot{x}_v^{(i)} = -1\right)(-2)e_h^{(i)} \quad (10)$$

Now on any example $i$, independent of the value of $e_h^{(i)}$, a $\rho$ fraction of the bits where $x_v^{(i)} = +1$ are flipped to get $\dot{x}_v^{(i)}$. Therefore,

$$\frac{1}{\rho}\sum_{i=1}^{n}\mathbf{1}\left(x_v^{(i)} = +1 \wedge \dot{x}_v^{(i)} = -1\right)e_h^{(i)} \approx \frac{1}{1-\rho}\sum_{i=1}^{n}\mathbf{1}\left(x_v^{(i)} = +1 \wedge \dot{x}_v^{(i)} = +1\right)e_h^{(i)}$$

Putting it all together,

$$\delta_{v,h} = \frac{1}{n}\sum_{i=1}^{n}(\dot{x}_v^{(i)} - x_v^{(i)})e_h^{(i)} = -\frac{2}{n}\sum_{i=1}^{n}\mathbf{1}\left(x_v^{(i)} = +1 \wedge \dot{x}_v^{(i)} = -1\right)e_h^{(i)}$$

$$\approx -\frac{2}{n}\frac{\rho}{1-\rho}\sum_{i=1}^{n}\mathbf{1}\left(x_v^{(i)} = +1 \wedge \dot{x}_v^{(i)} = +1\right)e_h^{(i)} = -\frac{2}{n}\frac{\rho}{1-\rho}\sum_{i=1}^{n}\mathbf{1}\left(\dot{x}_v^{(i)} = +1\right)e_h^{(i)}$$

# D   PROOFS

*Proof of Theorem 1.* Writing $\Gamma(\tilde{x}_v^{(i)}) := \ell_-(\tilde{x}_v^{(i)}) - \ell_+(\tilde{x}_v^{(i)}) = \ln\left(\frac{1+\tilde{x}_v^{(i)}}{1-\tilde{x}_v^{(i)}}\right)$ for convenience, we can simplify $\mathcal{L}^*$, using the definition of the loss (2), and Lagrange duality for all $VH$ constraints involving $\mathbf{B}$.

This leads to the following chain of equalities, where for brevity the constraint sets are sometimes omitted when clear, and we write $\mathbf{X}$ as shorthand for the data $\mathbf{x}^{(1)}, \ldots, \mathbf{x}^{(n)}$ and $\tilde{\mathbf{X}}$ analogously for the reconstructions.

$$\mathcal{L}^* = \frac{1}{2}\min_{\substack{\tilde{\mathbf{x}}^{(1)},\ldots,\tilde{\mathbf{x}}^{(n)} \\ \in[-1,1]^V}} \max_{\substack{\mathbf{x}^{(1)},\ldots,\mathbf{x}^{(n)}\in[-1,1]^V, \\ \forall v\in[V]: \frac{1}{n}\mathbf{E}\mathbf{x}_v=\mathbf{b}_v}} \frac{1}{n}\sum_{i=1}^{n}\sum_{v=1}^{V}\left[\left(1+x_v^{(i)}\right)\ell_+(\tilde{x}_v^{(i)}) + \left(1-x_v^{(i)}\right)\ell_-(\tilde{x}_v^{(i)})\right]$$

$$= \frac{1}{2}\min_{\tilde{\mathbf{X}}}\max_{\mathbf{X}}\min_{\mathbf{W}\in\mathbb{R}^{V\times H}}\left[\frac{1}{n}\sum_{i=1}^{n}\sum_{v=1}^{V}\left(\ell_+(\tilde{x}_v^{(i)}) + \ell_-(\tilde{x}_v^{(i)}) - x_v^{(i)}\Gamma(\tilde{x}_v^{(i)})\right) + \sum_{v=1}^{V}\mathbf{w}_v^\top\left(\frac{1}{n}\mathbf{E}\mathbf{x}_v - \mathbf{b}_v\right)\right]$$

$$\overset{(a)}{=} \frac{1}{2}\min_{\mathbf{w}_1,\ldots,\mathbf{w}_V}\left[-\sum_{v=1}^{V}\mathbf{b}_v^\top\mathbf{w}_v + \frac{1}{n}\min_{\tilde{\mathbf{X}}}\max_{\mathbf{X}}\sum_{v=1}^{V}\left[\sum_{i=1}^{n}\left(\ell_+(\tilde{x}_v^{(i)}) + \ell_-(\tilde{x}_v^{(i)}) - x_v^{(i)}\Gamma(\tilde{x}_v^{(i)})\right) + \mathbf{w}_v^\top\mathbf{E}\mathbf{x}_v\right]\right]$$

$$= \frac{1}{2}\min_{\mathbf{w}_1,\ldots,\mathbf{w}_V}\left[-\sum_{v=1}^{V}\mathbf{b}_v^\top\mathbf{w}_v + \frac{1}{n}\min_{\tilde{\mathbf{X}}}\sum_{i=1}^{n}\sum_{v=1}^{V}\left[\ell_+(\tilde{x}_v^{(i)}) + \ell_-(\tilde{x}_v^{(i)}) + \max_{\mathbf{x}^{(i)}\in[-1,1]^V}x_v^{(i)}\left(\mathbf{w}_v^\top\mathbf{e}^{(i)} - \Gamma(\tilde{x}_v^{(i)})\right)\right]\right]$$

$$(11)$$

where $(a)$ uses the minimax theorem (Cesa-Bianchi & Lugosi (2006)), which can be applied as in linear programming, because the objective function is linear in $\mathbf{x}^{(i)}$ and $\mathbf{w}_v$. Note that the weights are introduced merely as Lagrange parameters for the pairwise correlation constraints, not as model assumptions.

The strategy $\mathbf{x}^{(i)}$ which solves the inner maximization of (11) is to simply match signs with $\mathbf{w}_v^\top\mathbf{e}^{(i)} - \Gamma(\tilde{x}_v^{(i)})$ coordinate-wise for each $v \in [V]$. Substituting this into the above,

$$\mathcal{L}^* = \frac{1}{2}\min_{\mathbf{w}_1,\ldots,\mathbf{w}_V}\left[-\sum_{v=1}^{V}\mathbf{b}_v^\top\mathbf{w}_v + \frac{1}{n}\sum_{i=1}^{n}\min_{\tilde{\mathbf{x}}^{(i)}\in[-1,1]^V}\sum_{v=1}^{V}\left(\ell_+(\tilde{x}_v^{(i)}) + \ell_-(\tilde{x}_v^{(i)}) + \left|\mathbf{w}_v^\top\mathbf{e}^{(i)} - \Gamma(\tilde{x}_v^{(i)})\right|\right)\right]$$

$$= \frac{1}{2}\sum_{v=1}^{V}\min_{\mathbf{w}_v\in\mathbb{R}^H}\left[-\mathbf{b}_v^\top\mathbf{w}_v + \frac{1}{n}\sum_{i=1}^{n}\min_{\tilde{x}_v^{(i)}\in[-1,1]}\left(\ell_+(\tilde{x}_v^{(i)}) + \ell_-(\tilde{x}_v^{(i)}) + \left|\mathbf{w}_v^\top\mathbf{e}^{(i)} - \Gamma(\tilde{x}_v^{(i)})\right|\right)\right]$$

The absolute value breaks down into two cases, so the inner minimization's objective can be simplified:

$$\ell_+(\tilde{x}_v^{(i)}) + \ell_-(\tilde{x}_v^{(i)}) + \left|\mathbf{w}_v^\top \mathbf{e}^{(i)} - \Gamma(\tilde{x}_v^{(i)})\right| = \begin{cases} 2\ell_+(\tilde{x}_v^{(i)}) + \mathbf{w}_v^\top \mathbf{e}^{(i)} & \text{if } \mathbf{w}_v^\top \mathbf{e}^{(i)} \geq \Gamma(\tilde{x}_v^{(i)}) \\ 2\ell_-(\tilde{x}_v^{(i)}) - \mathbf{w}_v^\top \mathbf{e}^{(i)} & \text{if } \mathbf{w}_v^\top \mathbf{e}^{(i)} < \Gamma(\tilde{x}_v^{(i)}) \end{cases}$$

$$(12)$$

Suppose $\tilde{x}_v^{(i)}$ falls in the first case of (12), so that $\mathbf{w}_v^\top \mathbf{e}^{(i)} \geq \Gamma(\tilde{x}_v^{(i)})$. By definition of $\ell_+(\cdot)$, $2\ell_+(\tilde{x}_v^{(i)}) + \mathbf{w}_v^\top \mathbf{e}^{(i)}$ is decreasing in $\tilde{x}_v^{(i)}$, so it is minimized for the greatest $\tilde{x}_v^{(i)*} \leq 1$ s.t. $\Gamma(\tilde{x}_v^{(i)*}) \leq \mathbf{w}_v^\top \mathbf{e}^{(i)}$. This means $\Gamma(\tilde{x}_v^{(i)*}) = \mathbf{w}_v^\top \mathbf{e}^{(i)}$, so the minimand (12) is $\ell_+(\tilde{x}_v^{(i)*}) + \ell_-(\tilde{x}_v^{(i)*})$, where $\tilde{x}_v^{i*} = \frac{1-e^{-\mathbf{w}_v^\top \mathbf{e}^{(i)}}}{1+e^{-\mathbf{w}_v^\top \mathbf{e}^{(i)}}}$.

A precisely analogous argument holds if $\tilde{x}_v^{(i)}$ falls in the second case of (12), where $\mathbf{w}_v^\top \mathbf{e}^{(i)} < \Gamma(\tilde{x}_v^{(i)})$.

Putting the cases together, we have shown the form of the summand $\Psi$. We have also shown the dependence of $\tilde{x}_v^{(i)*}$ on $\mathbf{w}_v^{*\top} \mathbf{e}^{(i)}$, where $\mathbf{w}_v^*$ is the minimizer of the outer minimization of (11). This completes the proof. $\qquad\square$

## D.1 $L_\infty$ CORRELATION CONSTRAINTS AND $L_1$ WEIGHT REGULARIZATION

Here we formalize the discussion of Sec. 3.4 with the following result.

**Theorem 3.**

$$\min_{\tilde{\mathbf{x}}^{(1)},\dots,\tilde{\mathbf{x}}^{(n)}\in[-1,1]^V} \max_{\substack{\mathbf{x}^{(1)},\dots,\mathbf{x}^{(n)}\in[-1,1]^V, \\ \forall v\in[V]:\ \left\|\frac{1}{n}\mathbf{E}\mathbf{x}_v - \mathbf{b}_v\right\|_\infty \leq \epsilon_v}} \frac{1}{n}\sum_{i=1}^n \ell(\mathbf{x}^{(i)}, \tilde{\mathbf{x}}^{(i)})$$

$$= \frac{1}{2}\sum_{v=1}^V \min_{\mathbf{w}_v\in\mathbb{R}^H}\left[ -\mathbf{b}_v^\top \mathbf{w}_v + \frac{1}{n}\sum_{i=1}^n \Psi(\mathbf{w}_v^\top \mathbf{e}^{(i)}) + \epsilon_v\|\mathbf{w}_v\|_1 \right]$$

*For each $v, i$, the minimizing $\tilde{\mathbf{x}}_v^{(i)}$ is a logistic function of the encoding $\mathbf{e}^{(i)}$ with weights equal to the minimizing $\mathbf{w}_v^*$ above, exactly as in Theorem 1.*

*Proof.* The proof adapts the proof of Theorem 1, following the result on $L_1$ regularization in Balsubramani & Freund (2016) in a very straightforward way; we describe this here.

We break each $L_\infty$ constraint into two one-sided constraints for each $v$, i.e. $\frac{1}{n}\mathbf{E}\mathbf{x}_v - \mathbf{b}_v \leq \epsilon_v \mathbf{1}^n$ and $\frac{1}{n}\mathbf{E}\mathbf{x}_v - \mathbf{b}_v \geq -\epsilon_v \mathbf{1}^n$. These respectively give rise to two sets of Lagrange parameters $\lambda_v, \xi_v \geq \mathbf{0}^H$ for each $v$, replacing the unconstrained Lagrange parameters $\mathbf{w}_v \in \mathbb{R}^H$.

The conditions for the minimax theorem apply here just as in the proof of Theorem 1, so that (11) is replaced by

$$\frac{1}{2}\min_{\substack{\lambda_1,\dots,\lambda_V \\ \xi_1,\dots,\xi_V}}\left[ -\sum_{v=1}^V \left(\mathbf{b}_v^\top(\xi_v - \lambda_v) - \epsilon_v \mathbf{1}^\top(\xi_v + \lambda_v)\right) \right. \qquad (13)$$

$$\left. + \frac{1}{n}\min_{\tilde{\mathbf{X}}} \sum_{i=1}^n \sum_{v=1}^V \left[\ell_+(\tilde{x}_v^{(i)}) + \ell_-(\tilde{x}_v^{(i)}) + \max_{\mathbf{x}^{(i)}} x_v^{(i)}\left((\xi_v - \lambda_v)^\top \mathbf{e}^{(i)} - \Gamma(\tilde{x}_v^{(i)})\right)\right] \right]$$

$$(14)$$

Suppose for some $h \in [H]$ that $\xi_{v,h} > 0$ and $\lambda_{v,h} > 0$. Then subtracting $\min(\xi_{v,h}, \lambda_{v,h})$ from both does not affect the value $[\xi_v - \lambda_v]_h$, but always decreases $[\xi_v + \lambda_v]_h$, and therefore always decreases the objective function. Therefore, we can w.l.o.g. assume that $\forall h \in [H]: \min(\xi_{v,h}, \lambda_{v,h}) = 0$. Defining $\mathbf{w}_v = \xi_v - \lambda_v$ (so that $\xi_{v,h} = [w_{v,h}]_+$ and $\lambda_{v,h} = [w_{v,h}]_-$ for all $h$), we see that the term $\epsilon_v \mathbf{1}^\top(\xi_v + \lambda_v)$ in (13) can be replaced by $\epsilon_v \|\mathbf{w}_v\|_1$.

Proceeding as in the proof of Theorem 1 gives the result. $\qquad\square$

# E  GENERAL RECONSTRUCTION LOSSES

In this section we extend Theorem 1 to a larger class of reconstruction losses for binary autoencoding, of which cross-entropy loss is a special case. This uses techniques recently employed by Balsubramani & Freund (2016) for binary classification.

Since the data $\mathbf{X}$ are still randomized binary, we first broaden the definition of (2), rewritten here:

$$\ell(\mathbf{x}^{(i)}, \tilde{\mathbf{x}}^{(i)}) := \sum_{v=1}^{V} \left[ \left( \frac{1 + x_v^{(i)}}{2} \right) \ell_+(\tilde{x}_v^{(i)}) + \left( \frac{1 - x_v^{(i)}}{2} \right) \ell_-(\tilde{x}_v^{(i)}) \right] \tag{15}$$

We do this by redefining the partial losses $\ell_\pm(\tilde{x}_v^{(i)})$, to any functions satisfying the following monotonicity conditions.

**Assumption 1.** *Over the interval* $(-1, 1)$, $\ell_+(\cdot)$ *is decreasing and* $\ell_-(\cdot)$ *is increasing, and both are twice differentiable.*

Assumption 1 is a very natural one and includes many non-convex losses (see Balsubramani & Freund (2016) for a more detailed discussion, much of which applies bitwise here). This and the additive decomposability of (15) over the $V$ bits are the only assumptions we make on the reconstruction loss $\ell(\mathbf{x}^{(i)}, \tilde{\mathbf{x}}^{(i)})$. The latter decomposability assumption is often natural when the loss is a log-likelihood, where it is tantamount to conditional independence of the visible bits given the hidden ones.

Given such a reconstruction loss, define the increasing function $\Gamma(y) := \ell_-(y) - \ell_+(y) : [-1, 1] \mapsto \mathbb{R}$, for which there exists an increasing (pseudo)inverse $\Gamma^{-1}$. Using this we broaden the definition of the potential function $\Psi$ in terms of $\ell_\pm$:

$$\Psi(m) := \begin{cases} -m + 2\ell_-(-1) & \text{if } m \le \Gamma(-1) \\ \ell_+(\Gamma^{-1}(m)) + \ell_-(\Gamma^{-1}(m)) & \text{if } m \in (\Gamma(-1), \Gamma(1)) \\ m + 2\ell_+(1) & \text{if } m \ge \Gamma(1) \end{cases}$$

Then we may state the following result, describing the optimal decoding function for a general reconstruction loss.

**Theorem 4.** *Define the potential function*

$$\min_{\substack{\tilde{\mathbf{x}}^{(1)}, \ldots, \tilde{\mathbf{x}}^{(n)} \in [-1,1]^V}} \max_{\substack{\mathbf{x}^{(1)}, \ldots, \mathbf{x}^{(n)} \in [-1,1]^V, \\ \forall v \in [V]: \frac{1}{n}\mathbf{E}\mathbf{x}_v = \mathbf{b}_v}} \frac{1}{n} \sum_{i=1}^{n} \ell(\mathbf{x}^{(i)}, \tilde{\mathbf{x}}^{(i)})$$

$$= \frac{1}{2} \sum_{v=1}^{V} \min_{\mathbf{w}_v \in \mathbb{R}^H} \left[ -\mathbf{b}_v^\top \mathbf{w}_v + \frac{1}{n} \sum_{i=1}^{n} \Psi(\mathbf{w}_v^\top \mathbf{e}^{(i)}) \right]$$

*For each* $v \in [V], i \in [n]$, *the minimizing* $\tilde{\mathbf{x}}_v^{(i)}$ *is a sigmoid function of the encoding* $\mathbf{e}^{(i)}$ *with weights equal to the minimizing* $\mathbf{w}_v^*$ *above, as in Theorem 1. The sigmoid is defined as*

$$\tilde{\mathbf{x}}_v^{(i)*} := \begin{cases} -1 & \text{if } \mathbf{w}_v^{*\top} \mathbf{e}^{(i)} \le \Gamma(-1) \\ \Gamma^{-1}(\mathbf{w}_v^{*\top} \mathbf{e}^{(i)}) & \text{if } \mathbf{w}_v^{*\top} \mathbf{e}^{(i)} \in (\Gamma(-1), \Gamma(1)) \\ 1 & \text{if } \mathbf{w}_v^{*\top} \mathbf{e}^{(i)} \ge \Gamma(1) \end{cases} \tag{16}$$

The proof is nearly identical to that of the main theorem of Balsubramani & Freund (2016). That proof is essentially recapitulated here for each bit $v \in [V]$ due to the additive decomposability of the loss, through algebraic manipulations (and one application of the minimax theorem) identical to the proof of Theorem 1, but using the more general specifications of $\Psi$ and $\Gamma$ in this section. So we do not rewrite it in full here.

A notable special case of interest is the Hamming loss, for which $\ell_\pm(\tilde{x}_v^{(i)}) = \frac{1}{2} \left( 1 \mp \tilde{x}_v^{(i)} \right)$, where the reconstructions are allowed to be randomized binary values. In this case, we have $\Psi(m) = \max(|m|, 1)$, and the sigmoid used for each decoding neuron is the clipped linearity $\max(-1, \min(\mathbf{w}_v^{*\top} \mathbf{e}^{(i)}, 1))$.

# F    ALTERNATE APPROACHES

We made some technical choices in the derivation of PC-AE, which prompt possible alternatives not explored here for a variety of reasons. Recounting these choices gives more insight into our framework.

The output reconstructions could have restricted pairwise correlations, i.e. $\frac{1}{n}\tilde{\mathbf{X}}\mathbf{E}^\top = \mathbf{B}$. One option is to impose such restrictions *instead* of the existing constraints on $\mathbf{X}$, leaving $\mathbf{X}$ unrestricted. However, this is not in the spirit of this paper, because $\mathbf{B}$ is our means of indirectly conveying information to the decoder about how $\mathbf{X}$ is decoded.

Another option is to restrict both $\tilde{\mathbf{X}}$ and $\mathbf{X}$. This is possible and may be useful in propagating correlation information between layers of deeper architectures while learning, but its minimax solution does not have the conveniently clean structure of the PC-AE derivation.

In a similar vein, we could restrict $\mathbf{E}$ during the encoding phase, using $\mathbf{B}$ and $\mathbf{X}$. As $\mathbf{B}$ is changed only during this phase to better conform to the true data $\mathbf{X}$, this tactic fixes $\mathbf{B}$ during the optimization, which is not in the spirit of this paper's approach. It also performed significantly worse in our experiments.

