# Peer review of "Optimal Binary Autoencoding with Pairwise Correlations"

_ICLR 2017 — accepted_

[Official Review · AnonReviewer1 · rating 6 · confidence 4 · 16 Dec 2016]
**No Title**
originality 5 · substance 5

The paper propose to find an optimal decoder for binary data using a min-max decoder on the binary hypercube given a linear constraint on the correlation between the encoder and the  data. 
The paper gives finally that the optimal decoder as logistic of the lagragian W multiplying the encoding e.
 
Given the weights of the ‘min-max’decoder W the paper finds the best encoding for the data distribution considered, by minimizing that error as a function of the encoding.

The paper then alternates that optimization between the encoding and the min-max decoding, starting from random weights W.


clarity:

-The paper would be easier to follow if the real data (x in section 3 ) is differentiated from the worst case data played by the model (x in section 2). 


significance

Overall I like the paper, however I have some doubts on what the alternating optimization optimum ends up being.  The paper ends up implementing a single layer network. The correlation constraints while convenient in the derivation, is  a bit intriguing. Since linear relation between the encoding and the data  seems to be weak modeling constraint and might be not different from what PCA would implement.

- what is the performance of PCA on those tasks? one could you use a simple sign function to decode. This is related to one bit compressive sensing.

- what happens if you initialize W in algorithm one with PCA weights? or weighted pca weights?

- Have you tried on more complex datasets such as cifar?

[Official Review · AnonReviewer2 · rating 7 · confidence 2 · 17 Dec 2016]
clarity 4

The author attacks the problem of shallow binary autoencoders using a minmax game approach. The algorithm, though simple, appears to be very effective. The paper is well written and has sound analyses. Although the work does not extend to deep networks immediately, its connections with other popular minmax approaches (eg GANs) could be fruitful in the future.

[Official Review · AnonReviewer3 · rating 7 · confidence 3 · 19 Dec 2016]
soundness 2

The paper presents a novel look at binary auto-encoders, formulating the objective function as a min-max reconstruction error over a training set given the observed intermediate representations. The author shows that this formulation leads to a bi-convex problem that can be solved by alternating minimisation methods; this part is non-trivial and is the main contribution of the paper. Proof-of-concept experiments are performed, showing improvements for 1-hidden layer auto-encoders with respect to a vanilla approach. 

The experimental section is fairly weak because the literature on auto-encoders is huge and many variants were shown to perform better than straightforward approaches without being more complicated (e.g., denoising auto-encoders). Yet, the paper presents an analysis that leads to a new learning algorithm for an old problem, and is likely worth discussing.

[Final Decision · Program Chairs · 06 Feb 2017]
**ICLR committee final decision**

All reviewers (weakly) support the acceptance of the paper. I also think that binary neural networks model is an important direction to explore.